# Why Do Children with Acute Lymphoblastic Leukemia Fare Better Than Adults?

**DOI:** 10.3390/cancers13153886

**Published:** 2021-08-02

**Authors:** Alexandra Neaga, Laura Jimbu, Oana Mesaros, Madalina Bota, Diana Lazar, Simona Cainap, Cristina Blag, Mihnea Zdrenghea

**Affiliations:** 1Department of Hematology, Iuliu Hatieganu University of Medicine and Pharmacy, 400012 Cluj-Napoca, Romania; jimbulaura@yahoo.com (L.J.); mesaros.oana@gmail.com (O.M.); mzdrenghea@umfcluj.ro (M.Z.); 2Department of Pediatric Oncology and Hematology, Emergency Hospital for Children, 400177 Cluj-Napoca, Romania; lazardianaraluca@gmail.com (D.L.); cristinablag@yahoo.com (C.B.); 3Department of Hematology, Ion Chiricuta Oncology Institute, 400015 Cluj-Napoca, Romania; 4Department of Mother and Child, Iuliu Haţieganu University of Medicine and Pharmacy, 400177 Cluj-Napoca, Romania; bota.madalina@gmail.com (M.B.); cainap.simona@gmail.com (S.C.); 5Department of Pediatric Cardiology, Emergency Hospital for Children, 400177 Cluj-Napoca, Romania

**Keywords:** acute lymphoblastic leukemia, prognosis, survival rates, adolescents and young adults, differences in treatment response, pediatric-like protocols, novel therapies

## Abstract

**Simple Summary:**

Around 90% of children diagnosed with acute lymphoblastic leukemia are long-term survivors due to the efforts made in the last decades to tailor the chemotherapy protocols, which is not the case for the adult population. This literature review proposes to bring together all the relevant data to answer the ardent question: why the results in adults, adolescents and young adults are not nearly as good as those obtained in children?

**Abstract:**

It is a new and exciting time for acute lymphoblastic leukemia (ALL). While nearly 50 years ago, only one in nine children with ALL survived with chemotherapy, nowadays nearly 90% of children have a chance of long-term survival. Adults with ALL, as well as the special category of adolescents and young adult (AYA) patients, are catching up with the new developments seen in children, but still their prognosis is much worse. A plethora of factors are regarded as responsible for the differences in treatment response, such as age, ethnicity, disease biology, treatment regimens and toxicities, drug tolerance and resistance, minimal residual disease evaluation, hematopoietic stem cell transplantation timing and socio-economic factors. Taking these factors into account, bringing pediatric-like protocols to adult patient management and incorporating new agents into frontline treatment could be the key to improve the survival rates in adults and AYA.

## 1. Introduction

A crucial development in acute lymphoblastic leukemia (ALL) management occurred in 1947 when it was demonstrated that folic acid antagonists were efficient in inducing remission. The importance of acquiring complete remission (CR), the use of chemotherapeutic agents in combination, the prophylactic administration of intrathecal drugs, maintenance treatment during remission and the improvement of supportive care are important factors that have contributed to a gradual increase in survival [1]. In the history of hematological malignancies management, there are few achievements as spectacular as the massive decline in the mortality from childhood leukemia. Between the 1960s to the 2000s, progress led to an increase of children surviving ALL from one in nine to approximately nine in ten. This improvement is attributed not as much to the discovery of new drugs, as it is to the re-evaluation of the tools already in hand [2].

ALL can affect all individuals, from birth to the late stages of life, making it a very heterogenous disorder [3]. Over the last decades, patients with ALL have seen improved survival rates. However, as previously mentioned, this progress has mainly occurred in children and adolescents, with current 5 year event-free survival (EFS) rates varying from 76% to 86% [4,5] and getting up to 90% in some reports [6], whereas adults have much worse outcomes. ALL has a bimodal distribution regarding age, with peak incidences in children aged between 2 and 5 years and in adults older than 40 years [5]. In the pediatric population group, older children have poorer outcomes, and within adult populations, younger adults have better outcomes [7]. The adolescent and young adults (AYA) thus stand at the crossroads between these two age groups [8]. Survival rates in AYAs (generally defined as 16–39 years, but this definition is a subject of debate) are inferior and can decline by 50% between childhood and adulthood [9]. A recent analysis [10,11] indicated a better survival for adults in the past two decades, the most substantial improvement being seen in adolescents aged from 15 to 19 years, but still faring worse than children. The 5-year overall survival (OS) is 87% for children aged 0–15 years, as opposed to 63% for AYA aged 15–20 years and 44% for adults aged 20–29 years [12]. ALL is still a relatively infrequent disease in AYAs, so the data on this age group are limited and they are often analyzed either together with children or adults, depending on the pediatric or adult oncologists treating them [13]. The factors responsible for the different outcomes are various, including the disease heterogeneity, socio-economic factors, host factors, therapeutic protocols used and the experience of the health care professionals [14].

## 2. Socio-Economic Factors

ALL is the most common malignancy treated by pediatric hematologists [15]. Almost all children diagnosed with ALL are treated in specialized pediatric oncology centers, which have vast experience in this area, being involved in clinical trials, while adults are mostly treated by oncologists and probably have inferior access to clinical trials. Less than 2% of adolescents are enrolled in trials and this fact is associated with a poor outcome. They are treated by physicians and support teams with less experience with this disorder [12,15]. It is often invoked that young adults have poor compliance, are living independently, want to be emancipated, are concerned about fertility issues [16], are probably without parent support, which in children can help keeping a rigorous schedule of appointments or maintenance medication (given that this is administered orally). The greatest adherence to treatment is when patients are surrounded by a caregiver: a mother figure, relatives, loved ones, and a skilled team of professionals, which is often the case in pediatric patients [7]. The pediatric oncologists give more attention to the detail of therapy, are more determined to deliver treatment on time, avoiding delays based on nonmedical factors, administer the maximal dose and are more accustomed with protocols, chemotherapy-related toxicities and supportive care [6,8,17,18].

Another factor contributing to a worse outcome in young adults in certain countries could be the lack of proper medical insurance. In addition, comprehensive care centers are mostly dedicated to treating either younger or older patients, leaving them disconnected [19]. Racial disparities could also be an important factor due to the inequal access to health care systems and recent advancements in diagnosis and therapy [20].

There is a need for collaboration between centers both nationwide and internationally, because of the low number of AYA patients, which, as mentioned before, have less access to clinical trials, and there is a suboptimal accrual of cases to allow for statistical power [21].

## 3. Host Factors

### 3.1. Age

Age is a powerful determining factor for survival in ALL and the impact of age continues through adolescence, with a 10-year-old having the half of the risk of therapy failure of a 20-year-old, making age a constant variable [22]. For older patients, progress in the unfavorable outcome has been less significant, partly due to comorbidities and organ dysfunctions, making them unsuitable for intensive chemotherapy [23].

### 3.2. Sex and Ethnicity

Adverse factors such as male sex or African-American ethnicity lost their prognostic power with improved treatment strategies [24].

Another host factor that could be taken into consideration is sexual maturity, as sexual hormones might hypothetically influence the anti-leukemic drug metabolism [25].

### 3.3. Disease Biology

The difference in outcomes across age categories can be explained by numerous biological disparities. One of the significant differences between age categories is disease cytogenetics (Table 1). Also, adults usually present with a higher number of white blood cells, an increased frequency of T-cell ALL (20–25% in adults as compared to 15% in children), decreased incidence of hyperdiploidy and differences in sensitivity to corticosteroids and chemotherapy in vitro. Adolescents have intermediate characteristics between young children and adults and the frequency of T-cell phenotype is similar to adults and two times higher than in children [26]. However, children with T-cell ALL have a better prognosis compared to adults [27]. In T-ALL, inactivating PTEN lesions as well as NRAS/KRAS mutations can be found in children, but these do not influence prognosis, while in adults their presence is linked to a worse outcome [27,28]. Also, children often present with favorable trisomies of chromosomes 4, 10 or 17 [16,24].

The genomic landscape of adult ALL is different compared to children, as unfavorable prognostic cytogenetic anomalies increase with age. It is well known that the presence of the Philadelphia chromosome implies a poor prognosis. It has an incidence of only 3% in patients younger than 18 years of age but increases to 6% in patients up to 25 years, 14% in patients between 25–35 years and goes as high as 53% in patients older than 55 years [17].

The t(12;21) (ETV6/RUNX1) rearrangement is found in less than 3% in adults, whereas in children it is as high as 20% and it was observed that in adolescents the frequency was only 7% in the FRALLE-93 trial [26]. Favorable cytogenetics are less frequent in adults, but multiple analyses showed that even when present, they are associated with inferior survival [12].

The translocation t(4;11) (MLL) is commonly associated with infants (85% of the cases) but is also found in adults (3–8%) and they tend to be older and most frequently have a high leukocyte count, organomegaly and CNS involvement. Prognosis of MLL gene rearrangement is poor [38].

TCF3-PBX1 fusion t(1;19) is an interesting biomarker in ALL. A total of 3% of children and adolescents and 6% of adults carry this translocation and it correlates with pre-B-ALL, and it creates controversy regarding the risk once patients are diagnosed with this fusion. In pediatric cohorts, early studies regarded TCF3-PBX1 as a marker of poor prognosis, but more recent studies involving more intensive chemotherapy regimens have reported improved outcomes. Other studies reported CNS relapses and dismal outcomes after first relapse, pointing to clinical heterogeneity. With adults there is a similar case, with recent studies showing that they should be associated with intermediate risk. However, other study groups plead to stratify adults as high-risk and treat them more aggressively [32].

The incidence of extramedullary disease such as CNS involvement might be higher in AYA patients, and in children it is only 3% [16].

A significant number of studies concentrating their attention on extrapolating from pediatric to adult ALL acknowledged molecular mutations more frequently found with advancing age, such as IKZF1 gene deletions, JAK mutations or CRLF2 gene alterations/overexpression, immunoglobulin heavy chain (IGH) translocations, and iAMP21 [3,25,35,39,40,41]. IKZF1 deletion, a well-known high-risk feature in pediatric ALL, is connected to poor survival rates in adult ALL. Interestingly, IKZF1 deletion in patients without the ERG deletion is regularly combined with other gene deletions, such as CDKN2A/2B, PAX5 and PAR1, named the IKZK^plus^ gene abnormality [33]. IGH translocations are often found in AYAs and are linked to an unfavorable outcome in adults but are not an independent prognostic feature in children and adolescents [35]. Multiple studies indicated that the BCR-ABL-like gene expression profile is common with advancing age and is associated with a poor outcome [27].

Translocations of the double homebox 4 gene DUX4 into the IGH enhancer locus characterizes the DUX4/ERG cases. DUX4 is not normally expressed in B cells and translocations into the IGH leads to the expression of a truncated DUX4 isoform in the B-cell lineage. Intragenic deletions of the ERG gene have been found in 5% of childhood ALL and the dysregulated DUX4/ERG is connected to favorable outcomes, albeit the common presence of IKZF 1 intragenic deletions. DUX4/ERG is found in up to 15% of AYAs [30].

Early thymic precursor (ETP) ALL is a subgroup of T-cell ALL and was first described in children, having a neutral prognosis. However, in adults, the prognosis is dismal, with a high rate of treatment resistance when treated with traditional therapeutic protocols. ETP-ALL is thought to be more frequent in adults than in children and using RNA-sequencing profiling it was shown that adults had frequent mutations in DNA methylation factor genes, which were never found in children. This could demonstrate that adult ETP ALL could be genetically more similar to acute myeloblastic leukemia (AML) [27,42]. It was suggested, using pediatric-inspired protocols in adults (GRAAL 2003, GMAALL) that the disease resistance does not seem to be influenced by the different genetic ALL subtypes among the age groups, at least when patients receive intensive pediatric inspired treatment [25]. Also, in a study it was shown that conventional risk factors (WBC count, immunophenotype, early steroid resistance and some conventional cytogenetics, such as t (12;21)) may as well be abandoned as prognostic factors when using pediatric-inspired protocols, because it was found that oncogenetic events, such as MLL rearrangement, including t (4;11), BCR-ABL-like cases with IKZF1 deletions, CRFL2 alterations and early MRD assessment independently influenced the risk of relapse in patients enrolled in the GRAAL 2003/2005 trials. A comprehensive investigation of prognostic ALL markers is desired, including both conventional and complex cytogenetics and MRD assessment for optimal treatment stratification [25,43].

There is increasing evidence that adult ALL cells are more resistant to chemotherapy and have a higher MRD after therapy compared to children. In one experiment, ALL cells of children older than 10 years, which had identical cytogenetical features, were shown to be more resistant to chemotherapy in vitro compared to their counterparts of children younger than 10 years of age [3].

Although many recurrent chromosomal aberrations could explain the difference in outcome, they are not enough to account for the variations in response to therapy. In the context of recent advances and availability of genomics, many groups performed gene profiling, which demonstrated different gene profiles related to recurrent chromosomal abnormalities and also identified novel aberrations in ALL [12].

With increasing age, the mechanisms involved in ALL pathogenesis are becoming more complex regarding the pathways and different target cells involved in malignant transformation, which can partly explain the dismal prognosis in adult ALL. The latest studies of clonal development of pediatric and adult ALL found enhancement of mutations in epigenetic regulators from diagnosis to relapse and it is feasible that epigenetic modifier abnormalities could lead to chromatin changes, which could determine a greater drug resistance in adult B cell ALL. Adults with ALL have more cooperative alterations and mutations of epigenetic modifiers and genes linked to B-cell development, indicating differences in the transformation of target cells between adult and pediatric patients, explaining the differences in the treatment response [44].

Many other studies have tried to find new biomarkers which can predict the clinical outcome in adult ALL, but the majority used only one or two markers at a time and because ALL is heterogeneous, their prognostic value is poor. The hypothesis is that leukemic cells have a high turnover rate, and they release into circulation proteins and DNA, which could function as biomarkers forecasting recurrence and these data could be used for a proteomic approach. In the future, proteomic analyses could be relevant by identifying useful biomarkers for the characterization of ALL and the prediction of disease progression [45].

### 3.4. Treatment Related Differences

Almost two decades ago, it became clear that the gap in outcome between children and adults could not be attributed solely to differences in disease biology and treatment tolerance, and that improved response rates and prolonged survival could be associated with the intensified chemotherapy regimens in children. An important question is why treatment strategies differ insofar as to sometimes be opposed. It is known that adult hematologists prefer chemotherapy regimens that resemble AML treatment, based on induction and short consolidation blocks, followed closely by hematopoietic stem cell transplantation (HSCT) [25].

Over the past years, the treatment of adults diagnosed with ALL has significantly improved. Most study groups now use pediatric-inspired regimens or even unmodified pediatric protocols in adults up to 60 years old, so chemotherapy intensity has increased [43]. In a comparison of the pediatric FRALLE-93 trial and adult LALA-94 trial, it was observed that adolescents treated with the pediatric protocol were stratified in the high-risk group, thus receiving more intense chemotherapy, whereas the LALA-94 considered them as standard risk group patients. Better results were obtained for adolescents treated with the pediatric protocol regarding the complete remission rate achieved and event-free survival rate (EFS) [26].

There is a significant difference in induction courses between pediatric and adult protocols, meaning that usually children protocols administer higher doses of non-myelotoxic agents, like vincristine, steroids and L-asparaginase, higher doses of methotrexate within a shorter interval of time, the time to recovery is shorter, and continuously higher doses of prednisone and more asparaginase are used, leading to a better outcome (Table 2). On the contrary, adult protocols administer higher doses of anthracyclines, cytarabine, cyclophosphamide and etoposide, particularly in AML-like consolidation cycles. The combination of high-dose cytarabine and mitoxantrone causes a much-prolonged period of neutropenia and thrombocytopenia, thus not allowing a short interval between chemotherapy cycles [25,26]. Commonly, adult protocols have longer delays between courses and pediatric hematologists administer chemotherapy with greater adherence to schedules and dose intensity [19].

Several studies in young populations in the USA, France, Netherlands and the UK concluded that adults treated with pediatric regimens have superior outcomes as compared to those treated with classical adult regimens [13,18,22,26,46]. These differences cannot only be accounted for by the median age or disease biology, as children and AYAs have comparable rates of Ph+ and MLL rearrangements and other high-risk characteristics. It is likely that many AYAs could be underdosed when receiving treatment based on adult protocols, due to insufficient total doses of corticosteroids, vinca alkaloids and asparaginase. AYAs and adults up to 50 years old can tolerate pediatric regimens and this can lead to improved outcomes [18,53,54,55].

Most clinical trials in adults to date included patients between 15 and 80 years. A small decrease in complete remission rates (CR) was observed with advancing age, but overall survival rates (OS) were far better in younger patients as compared to older patients treated with the same pediatric-inspired regimen. Even though better results were obtained in younger patients treated with adult protocols, it is possible that these younger patients could be getting less intensive chemotherapy and therefore be undertreated [19].

In addition to these differences, children protocols consider that early and more intensive CNS therapy is a standard of care, compared to adult regimens, and more prolonged maintenance chemotherapy is considered essential [18].

In children with Ph + ALL who receive intensive chemotherapy and a tyrosine kinase inhibitor, cure rates of up to 70% can be achieved, as opposed to adults receiving the same treatment, in which cure rates of less than 50% are obtained, even with the addition of transplantation [44].

A single dose intensive strategy is insufficient to result in substantial survival benefits in adults and room for improvement remains. The heterogeneity of ALL in adults indicates that improved outcomes could be obtained by incorporating targeted therapy into frontline treatment such as nelarabine, clofarabine, rituximab, the antibody-drug conjugate inotuzumab ozogamicin (anti-CD22 bound to the antitumor antibiotic calicheamicin), blinatumomab (a bispecific CD3 anti CD19 T cell engager, that links and directs endogenous CD3 T cells against CD19 B cells, inducing apoptosis-BiTE), the first and only FDA-approved BiTE, chimeric antigen receptor T cells and offering enrollment into clinical trials [14,56,57,58,59]. However, the duration of response achieved alone with these novel agents is dismal [47] (Figure 1).

### 3.5. Treatment Related Toxicities

Pediatric oncologists are more accustomed with drug-related toxicities compared to adult oncologists, being more prone to maintain the recommended dose and schedule. In the FRALLE-93 trial, the time interval between achieving complete remission (CR) and the next course of chemotherapy was significantly shorter when compared to the adult LALA-94 trial (2 versus 7 days) [17].

Older ALL patients can have lower rates of complete remission and more treatment-related toxicities due to decreased drug tolerance (such as asparaginase), resistance to treatment agents (corticosteroids, L-asparaginase, cytarabine, daunorubicin, vincristine) and alterations in drug metabolism [46].

One of the main questions of using a pediatric inspired-regimen in AYA is that of asparaginase-related toxicities. In a recent study which enrolled adults aged 18–50 years old, who received an intensive pediatric protocol with 30 weeks of high-dose asparaginase, it was found that adults who completed 26 or more weeks of this drug had similar treatment efficacy with the pediatric population. Tolerance to asparaginase is linked to a favorable prognosis and the occurrence of asparaginase-induced toxicities (hepatic toxicity, hypersensitivity reactions, neutralizing antibodies, pancreatitis, thrombosis, bleeding) was comparable to that found in older children. However, tolerance to asparaginase of adults was slightly lower than in children [3,18].

A recent study demonstrated that blinatumomab, besides inducing remission and allowing for the bridging to HSCT, can also provide effective therapy during severe infections until chemotherapy can be resumed [59]. In the phase III TOWER study, notable improvements were noted in adults with relapsed or refractory ALL treated with blinatumomab, irrespective of age, prior treatment, prior HSCT, or bone marrow disease burden, but it was more evident in the first salvage. Blinatumomab is now approved to treat MRD-positive patients [49]. In another study on patients aged 1–30 years with intermediate-risk or high-risk relapsed ALL which were randomized in two study arms, one arm receiving two blocks of intensive treatment and the other re-induction followed by two 4-week blocks of blinatumomab proved that the second arm had improved 2-year overall survival and MRD negativity with lower rates of febrile neutropenia, infection and sepsis. Also, in one study, adult patients with relapsed/refractory ALL receiving inotuzumab ozogamicin achieved better rates of remission than those with conventional treatment. In children, however, inotuzumab ozogamicin is connected to high rates of veno-occlusive disease, especially after HSCT, and it was observed that if lower doses, fractioned inotuzumab, and a prolonged interval to HSCT were used, this complication could be avoided [48]. The revolutionizing CAR-T cell treatment of patients with relapsed/refractory B-ALL has yielded high response rates, but with short durations, especially in adult populations. Another issue to be addressed is the toxicity management, such as prolonged aplasia, hypogammaglobulinemia, cytokine release syndrome, and immune effector cell-associated neurotoxicity syndrome, which are major problems [47].

### 3.6. Drug Tolerance and Drug Resistance

Most of the cytostatic drugs used nowadays have been known for over four decades, but it is less known what their safest antileukemic dose is, the best administration schedule or the differences between each patient’s drug metabolism [7].

Using intensive unmodified pediatric protocols in adult ALL patients could lead to more adverse reactions; for example, high doses of prednisone can cause hypertension or hyperglycemia, there can be a more prolonged myelosuppression, higher incidence of vincristine- or asparaginase-related toxicities or late results such as secondary malignancies induced by therapy [25]. This radical option is feasible in teenagers and young adults, who perhaps can tolerate this approach as well as children, but it is suggested that in adults, pediatric-inspired regimens should be used [53].

Pediatric ALL clinical trials have demonstrated that dexamethasone is far superior to prednisone, largely by decreasing the risk of CNS relapse [19]. It was observed that in vitro resistance to prednisolone with increasing age might be a continuous variable in ALL patients, except infants. This can be attributed to the activation of P-glycoprotein, which has a higher expression with advancing age, lower methotrexate polyglutamate accumulation and perhaps mutations in the p53 gene in adults [60]. Protein tyrosine phosphatase nonreceptor type 2 (PTPN2) is known for its function for suppressing a gene in T-cell ALL. PTPN2 deletions are associated with αβ lineage and TLX deregulation and a positive relationship with alterations in the IL7R/JAK/STAT signaling pathway. These deletions are also connected to a higher glucocorticoid response and better survival rates in children, but not in adults [50]. It was shown in a study that a complex karyotype (≥3 cytogenetic alterations) could in part explain the steroid resistance associated with activating mutations in IL7R in adults with T-ALL. A recent report suggested that these patients were slow responders with a high burden MRD on the 8th day of treatment, despite no correlation being found between the two groups regarding the prednisone response [51]. PRC2 (Polycomb Repressor Complex 2) loss-of-function alterations were found in pediatric T-ALL. In a study of poor prednisone poor response, low bone marrow blast clearance and persistent MRD in T-ALL adult patients were connected factors. PRC2 function loss intertwines with activating mutations of the IL7R/JAK/STAT pathway and are common mutations in T-ALL but are not restricted to ETP-ALL [52].

The Cancer and Leukemia Group B (CALGB) study demonstrated that the administration of L-asparaginase is well tolerated by adult patients and that patients with T-cell ALL had a better prognosis compared to those with pre-B-cell ALL, particularly if they had mediastinal masses [19].

The France–Belgium Group for Lymphoblastic Leukemia in Adults 94 (LALA-94 trial) suggested that pediatric-inspired regimens have substantially improved patient outcomes, although a worse treatment tolerance was noted in patients older than 45 years. In the GRAALL 2003 study, the chemotherapy toxicity was satisfactory in adults younger than 45 years, but older patients did not tolerate induction or postremission therapy. Even if they still benefitted from this approach, the cumulative rate of chemotherapy-related deaths was still too high (23%). There is a necessity of dose adaptations and reduced intensity conditioning for HSCT in these older patients [53].

### 3.7. Time to Complete Remission

Children have a tendency to achieve complete remission earlier than adult patients, confirmed by the MRD negativity [16].

## 4. MRD Evaluation

Obtaining an early remission, prevention of relapse and treatment-associated mortality are important therapeutic steps. Evaluation of the disease response to the first phase of intensive chemotherapy through minimal residual disease (MRD) monitoring is essential and can impact prognosis, risk group stratification and treatment approach, being the strongest predictor of relapse [61,62,63]. In the pediatric population, MRD evaluation is routinely used, and a major point is that the purpose of chemotherapy is achieving MRD negativity. In adults, however, MRD evaluation has only recently been incorporated into treatment algorithm. The information that MRD brings is as important as the initial white blood count or cytogenetics [28,34,64]. Using the MRD response can accurately select patients for HSCT, thus sparing adult patients with negative MRD from transplant-related toxicities. A total of 70–80% of patients stratified in the standard risk group and 50% of high-risk group patients could achieve negative MRD, making it the most important prognostic factor known in adult ALL, particularly in young adults treated with novel pediatric-like regimens. Relapse can be expected from MRD positivity, and the risk of relapse is much higher in standard risk group patients with positive MRD than the high-risk group ones, thus the prognosis is dismal on chemotherapy alone [61,65,66]. Early MRD detection is an essential tool to assess the risk group stratification that should be used in adults treated with modern therapies; in fact, this has already been stated in six other studies published between 2000 and 2013, including more than 1000 patients and in almost all of these studies MRD was recognized as a strong predictor of outcome [43]. The MRD status is crucial in risk stratification, established by European and US pediatric studies. These studies use risk stratification based on MRD status postinduction or postconsolidation, increasing or decreasing therapy intensity. MRD status in adult patients could play a similar role as in the pediatric studies. A two-stage risk-adapted analysis found that 72% of patients with negative MRD were free of disease after 5 years, as opposed to only 14% of MRD positive patients, despite the stratification into the risk groups. This meta-analysis found that pediatric MRD patients had better chances to be disease-free after 10 years compared to those who were MRD positive (73% versus 32%). A total of 64% of MRD negative adults were free of disease after 10 years and only 21% of MRD positive were disease-free [67]. Although MRD is a strong tool at hand in the short term, in the long run it is a poor instrument for treatment effect at the trial level, with the necessity of randomized trials to carefully discern the MRD limitations regarding the long-term effects [62,68].

## 5. HSCT in First Complete Remission

Despite the many advances in ALL, cooperative group trials continue to identify subgroups of ALL with a higher risk for relapse following chemotherapy, who could benefit from HSCT. Improvements have been made in donor selection, conditioning regimens, immunosuppression, infection monitoring and prophylaxis, leading to the diminishing rate of transplant-related mortality (TRM). However, this risk is greater in AYAs and patients older than 13 years of age receiving HSCT and they were observed as having inferior outcomes due to two times greater TRM. Contrary to this finding, the relapse rate in children and AYA following a myeloablative conditioning regimen is similar and regardless of the patient’s age [69].

The tendency in adults is in fact to receive early HSCT, probably before an ideal decrease in MRD level. The usual risk factors that most European adult ALL groups have used to stratify patients in the high-risk group include high white blood cell count, immunophenotypic characteristics, cytogenetic features, karyotype, early response to therapy (MRD evaluation), and usually one feature is enough to consider a patient as being part of the high-risk group and to offer him HSCT in first CR if he has a donor. It was shown that patients with poor MRD response substantially benefited from HSCT in first CR and poor MRD response was a powerful prognostic factor and a powerful predictive element for a positive HSCT effect [25].

The most suitable therapeutic methods for ALL continue to evolve. HSCT in CR1 is the standard of care for both children and adults with high-risk characteristics. Specific genomic studies, pharmacogenomics and better MRD evaluation can improve the identification of candidates for HSCT in CR1 who are otherwise considered as a standard risk group. The presence of the Philadelphia chromosome is a clear indication of HSCT, but it should be also considered for patients with other adverse cytogenetics, high white blood cell count and MRD positivity [4]. When HSCT is incorporated in the treatment strategy in CR1 there is only a small improvement in OS, especially for patients in the standard risk group and TRM is an important issue in up to 40%. These findings were confirmed in a study performed on Australian ALL patients which included adolescents and adults treated on the FRALLE-93 protocol and it was shown also that high-risk patients had a poor prognosis with or without HSCT [6].

Whether adults can benefit from HSCT with a reduced intensity regimen remains an important question. A retrospective study from the European Society for Blood and Marrow Transplant concluded that serious consideration is to be given to a reduced intensity regimen for older adults with ALL due to a higher TRM [70].

Pediatric-inspired regimens can offer the same leukemia-free survival in the absence of HSCT. The BFM group has shown that for children with ALL and HSCT from a sibling donor, TRM was as low as 4%, leading to the observation that young adults tolerate HSCT more poorly than children. Further, after resolving the TRM, this method did not seem to decrease the risk of relapse. Patients older than 30 years had an excess of TRM, mostly due to comorbidities and deficient supportive care, emphasizing the fact that an optimal conditioning regimen is vital [71].

Achievement of satisfactory low or negative MRD with standard chemotherapy to proceed to HSCT is difficult, so intensive treatments come together with severe infections and end-organ consequences, further underlying the need for alternate therapeutic approaches. Blinatumomab proved to be efficient as a bridge to HSCT, especially in patients with positive MRD, in improving survival, both in children and AYA. In the BLAST study it was shown that adults receiving blinatumomab treatment prior to HSCT could be beneficial in selected patients, but long-term survival without HSCT could be also possible [59,63,72]. Inotuzumab ozogamicin, approved for use in adults in the relapse/refractory ALL and Ph + ALL setting, with achievement of high rates of negative MRD, is also used for transplantation bridging with satisfactory results. The best approach regarding the timing and the succession of antibody-based and cellular immunotherapies is still not completely established and further clinical trials will explore these agents [59,73]. It is not exactly known where CAR-T cell therapy stands in the HSCT setting. It has immunomodulatory effects, and it is associated with cytokine release syndrome with an impairment effect on the endothelium, which can affect the safety profile of HSCT after CAR-T therapy. Lymphodepletion prior to CAR-T infusion can have a harmful effect with an excess of morbidity and mortality [47].

## 6. Future Directions in Treating ALL

In contrast to the significant advances made in the standard treatment of pediatric and adult ALL, relapse and resistance to chemotherapy rates are still high, the doses of conventional chemotherapy are stretched to the limits, especially in the adult population, and the outcome is dismal. Small patients with ETV6-RUNX1 or hyperdiploidy (>50 chromosomes) with negative MRD during induction are proper candidates for treatment reduction. Antibody-based treatments are a major breakthrough and various mechanisms have been applied to target surface antigens that are usually expressed on the surface of blast cells, leading to the development of monoclonal antibodies (MAbs) [29,74]. Such MAbs are targeting CD20 (Rituximab), are antibody–drug conjugates targeting CD22 (inotuzumab ozogamicin), bispecific antibodies (Blinatumomab) and CD19 chimeric antigen receptor T cell therapy (tisagenlecleucel—the first CAR-T cell-based product approved by the FDA in august 2017 for relapsed/refractory ALL for children and AYA up to 25 years old). The treatment of ALL was also revolutionized with the introduction of tyrosine-kinase inhibitors targeting fusion proteins (BCR-ABL1) [47,75,76,77]. The first major discovery in the use of monoclonal antibodies was made in 1997 with the approval of Rituximab (anti-CD20) by the FDA for non-Hodgkin lymphoma, but its use has extended nowadays [78]. Developments in the recombinant DNA technology headed to the progress of chimeric or humanized MAbs with reduced immunogenicity [77].

In the ALL setting there are thought to be many fusions or mutations that can serve as alleged targets, but efficient specific therapy for each target has yet to be established. Ph-positive patients or Ph-like ALL with ABL class fusion are suitable for therapy with Dasatinib. The identification of high-risk subtypes (hypodiploidy, ETP-ALL, rearrangements of KMT2A, immature T-cell, Ph positive, TCF-HLF) may respond to a bcl-2 inhibitor, venetoclax [74]. The activation of the Ras pathway is a usual finding in pediatric ALL and patients could benefit from RAF and MEK inhibitors, but there are no FDA-approved agents yet for such targets [79]. For those who do not have a targetable lesion or who do not respond early to chemotherapy, the options include blinatumomab, inotuzumab or CAR-T cells for B-ALL or nelarabine and daratumumab (antiCD38) for T-ALL [49,74].

Besides the well-known T-cell redirecting antibodies, there are other such bispecific IgM antibodies designed to bind to one or two tumor-associated antigen molecules (IGM-2323 which has 10 binding units to CD20) to CD3. These molecules are useful in patients in which the expression of CD20 has been reduced due to the treatment with other anti-CD20 antibodies, such as Rituximab. IGM2323 seems to produce less cytokines and a milder cytokine release syndrome, making it a promising therapeutic option for the future, although further studies are awaited. Tri-specific antibodies are being developed, which target cancer cells, receptors that activate T cells and co-stimulatory signals that enhance continuing T cell activity against malignant cells. Such antibodies resemble blinatumomab with the addition of a co-stimulatory domain that activates T cells, which at the same time targets CD3 and CD28 molecules. This method of co-stimulatory domains is well-known in CAR-T therapy design [48,49].

Other issues are addressed regarding the long periods of administration of blinatumomab (4 week-long perfusions) to maintain the therapeutic serum concentration, thus dual-affinity re-targeting proteins (DARTs) and tandem diabodies (TandAbs) have been created [49]. CAR-T therapy symbolizes a major development in treating hematological malignancies and efforts are being concentrated into increasing the potency, the persistence, decreasing toxicities, exploring new cancer-associated antigens as targets, implementing multi-targeted CARS, decreasing the long manufacturing process and decreasing the costs [47].

Using new techniques of detecting MRD positive patients, such as next generation sequencing (NGS) to identify specific genomic lesions that can be overlooked by the current methods, has a tremendous potential for making more efficient drug improvement by corroborating prompt evidence of treatment benefit [62,67,80]. Widespread sequencing and genome-wide analyses have distinguished new elements with prognostic and therapeutic consequences. Learning how the different pathways of genetic alterations function is essential for therapeutic targeting. Furthermore, functional genomic screening and testing for chemotherapeutic drug sensitivity broadens the horizon into individualized medicine, together with personalized targeted therapy, IMMUNOTHERAPY, reduced-intensity chemotherapy, or even without chemotherapy [48,81]. The knowledge and the therapeutic options at hand are rapidly expanding and the efforts will concentrate on the balanced combinations of targeted therapies trying to avoid and to eliminate toxicities [74].

## 7. Conclusions

Over the last years, the outcome of adults diagnosed with ALL has improved, benefitting from more precise molecular diagnostic techniques, prompter and increasingly available MRD evaluation, and, not least, the intensification of therapy, the tendency nowadays being to follow pediatric therapeutic schemes. Adult ALL management is rapidly catching up with that of pediatric ALL. It is important to emphasize the fact that adolescents are not “big children” and to seek a better adaptation of therapy to their unique needs. The important question about the upper age limit up to which a pediatric protocol could be administered still remains unanswered. Another important question is to be addressed: which patients benefit more from HCST in CR1 and in which patients should this be withheld, warranting future investigation. Rather than intensifying doses in adults, new targeted therapy should be added to frontline chemotherapy, as novel agents are being developed with unprecedented celerity, having the potential to dramatically improve patient outcomes.

## Figures and Tables

**Figure 1 cancers-13-03886-f001:**
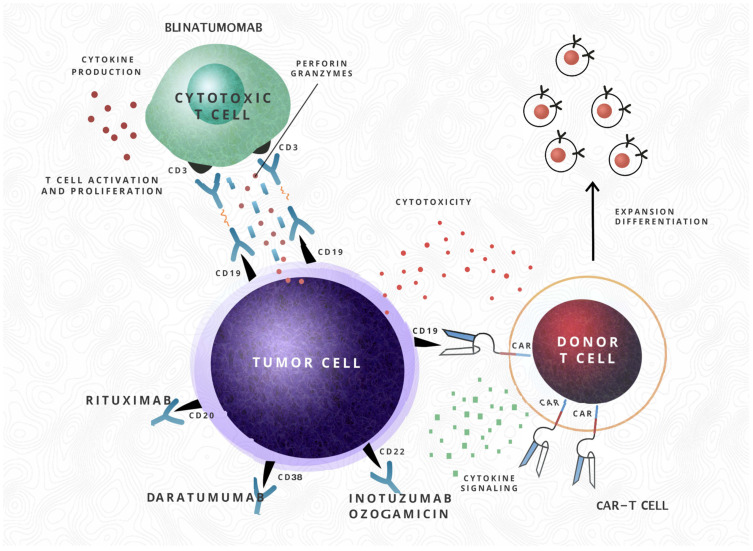
The schematic mechanisms of action of novel therapies. The mechanism of action of monoclonal antibody Rituximab (antiCD20), Daratumumab (antiCD38), antibody–drug conjugate Inotuzumab ozogamicin (antiCD22), bi-specific antibody Blinatumomab (CD3 antiCD19 T cell engager) and CAR-T cells (followed by expansion and differentiation after infusion).

**Table 1 cancers-13-03886-t001:** Frequency of disease characteristics of AYA, adults and children with ALL. AYAs were defined between 15 to 39 years of age. Ph-like prevalence is controversial between studies, being different in US and Europe cohorts. Some data are lacking, with AYAs having prevalence between children and adults.

Disease Characteristic	Adults	AYA	Children	References
High WBC count	More frequent	More frequent	Less frequent	[26,29]
T-cell	25%	Intermediate	15%	[30]
ETP-ALL	7.4%	-	10–15%	[12,31]
Hyperdiploidy	5%	Less than 20%	30–40%	[12]
Trisomies of chromosomes 4, 10, 17	Rare	Rare	Frequent	[16,32]
Philadelphia chromosome	53%	14%	3%	[17]
t (12;21)/ETV6/RUNX1	2%	7%	25%	[12]
IKZF1 gene deletions	20.3%	-	15%	[32,33]
IKZF1^plus^	21.3%	-	6%	[33]
Ph-like mutations	27%	25%	3%	[3,34]
JAK mutations	5%	60%	5.6%	[31]
CRLF2 gene alterations	4%	11%	5–7%	[12]
iAMP21	12%	5.8%	1.5%	[30]
IGH translocations	More frequent	11%	<3%	[32,35]
DUX4/ERG	~2%	15%	5%	[30,36]
t (4;11)/MLL	8–10%	4.5–5.7	2–3% (85% in infants)	[32,37]
t (1;19)/TCF3-PBX1	6%	3%	3%	[12,32]
CNS involvement	Higher	10%	3%	[29]

**Table 2 cancers-13-03886-t002:** Treatment differences between adults and children with ALL. Abbreviations: VCR-vincristine, MTX-methotrexate, CNS-central nervous system, CRS-cytokine release syndrome, PPR-poor prednisone response.

Characteristic	Adults	Children	References
**Chemotherapy**	Myelotoxic agents (anthracyclines, cytarabine, cyclophosphamide, etoposide)	Non-myelotoxic agents (VCR, asparaginase, MTX, higher doses of Prednisone)	[25,26]
	Longer delays between courses	Greater adherence to schedules	
		Early and more intensive CNS chemotherapy	[18]
		More prolonged maintenance chemotherapy	
**Increased frequency of drug toxicity**	Asparaginase hypersensitivity reactions, asparaginase, corticosteroids, cytarabine, daunorubicin, VCR toxicities Blinatumomab CRS CAR-T cells CRS, aplasia, hypogammaglobulinemia, immune effector cell-associated neurotoxicity syndrome	Inotuzumab ozogamicin (higher rates of veno-oclusive disease) Blinatumomab CRS CAR-T cells CRS, aplasia, hypogammaglobulinemia, immune effector cell-associated neurotoxicity syndrome)	[3,14,46,47,48,49]
	Alterations in drug metabolism		
**Drug resistance**	Resistance to prednisolone with increasing age		[50]
	Complex karyotype associated with steroid resistance in T-ALL		[51]
	PRC2 loss-of-function alterations associated with PPR in T-ALL		[52]
**Time to complete remission**		Children achieve CR earlier than most adults	[16]

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
