# Peer review of "Why Do Children with Acute Lymphoblastic Leukemia Fare Better Than Adults?"

_cancers, 2021, doi:10.3390/cancers13153886_

Round 1

Reviewer 1 Report

The authors claim to gather all relevant data in order to answer the question why the outcome of ALL in adults and adolescents and young adults is not as good as that in children. This review displays a convenient and useful comparison between age groups, attending to different parameters. In my opinion, it lacks however some thoroughness for it to be a complete guide to consult.

Specific comments:

  • Table 1 contains exactly the same information as figure 2 in (doi: 10.3324/haematol.2015.141101), thus adding no significant knowledge to the field.
  • Table 2 lacks some information that could be have been obtained from the literature, like JAK alterations in AYAs (doi:10.1056/NEJMoa1403088) or IKZF1 implication in AYAs (doi: 10.1200/JCO.2017.76.7228). Moreover, this table would need an additional column with the referred articles for each row.
  • Additionally, I miss some other table(s) gathering the information for sections related to treatment (differences, toxicities, response / resistance, etc).
  • The bibliography does not include relevant recent references, just as an example:

1: Fang Q, Song Y, Gong X, Wang J, Li Q, Liu K, Feng Y, Hao Q, Li Y, Wei H, Zhang G, Liu Y, Gong B, Wang Y, Zhou C, Lin D, Liu B, Wei S, Gu R, Mi Y, Wang J. Gene Deletions and Prognostic Values in B-Linage Acute Lymphoblastic Leukemia. Front Oncol. 2021 Jun 2;11:677034. doi: 10.3389/fonc.2021.677034. PMID: 34150641; PMCID: PMC8206559.

2: Genescà E, Morgades M, González-Gil C, Fuster-Tormo F, Haferlach C, Meggendorfer M, Montesinos P, Barba P, Gil C, Coll R, Moreno MJ, Martínez- Carballeira D, García-Cadenas I, Vives S, Ribera J, González-Campos J, Díaz-Beya M, Mercadal S, Artola MT, Cladera A, Tormo M, Bermúdez A, Vall-Llovera F, Martínez-Sánchez P, Amigo ML, Monsalvo S, Novo A, Cervera M, García-Guiñon A, Ciudad J, Cervera J, Hernández-Rivas JM, Granada I, Haferlach T, Orfao A, Solé F, Ribera JM. Adverse prognostic impact of complex karyotype (≥3 cytogenetic alterations) in adult T-cell acute lymphoblastic leukemia (T-ALL). Leuk Res. 2021 Jun 8;109:106612. doi: 10.1016/j.leukres.2021.106612. Epub ahead of print. PMID: 34139642.

3: Cristiano G, Nanni J, Sartor C, Parisi S, Marconi G, Barbato F, Arpinati M, Bonifazi F, Curti A, Cavo M, Paolini S, Papayannidis C. Clinical Efficacy of Ponatinib in Philadelphia-Positive T-Cell Acute Lymphoblastic Leukemia with Extramedullary Involvement. Acta Haematol. 2021 Jun 15:1-5. doi: 10.1159/000516003. Epub ahead of print. PMID: 34130278.

4: Andrieu GP, Kohn M, Simonin M, Smith C, Cieslak A, Dourthe ME, Charbonnier G, Graux C, Rigal-Huguet F, Lheritier V, Dombret H, Spicuglia S, Rousselot P, Boissel N, Asnafi V. PRC2 loss of function confers a targetable vulnerability to BET proteins in T-ALL. Blood. 2021 Jun 14:blood.2020010081. doi: 10.1182/blood.2020010081. Epub ahead of print. PMID: 34125178.

5: Shah BD, Ghobadi A, Oluwole OO, Logan AC, Boissel N, Cassaday RD, Leguay T, Bishop MR, Topp MS, Tzachanis D, O'Dwyer KM, Arellano ML, Lin Y, Baer MR, Schiller GJ, Park JH, Subklewe M, Abedi M, Minnema MC, Wierda WG, DeAngelo DJ, Stiff P, Jeyakumar D, Feng C, Dong J, Shen T, Milletti F, Rossi JM, Vezan R, Masouleh BK, Houot R. KTE-X19 for relapsed or refractory adult B-cell acute lymphoblastic leukaemia: phase 2 results of the single-arm, open-label, multicentre ZUMA-3 study. Lancet. 2021 Jun 3:S0140-6736(21)01222-8. doi: 10.1016/S0140-6736(21)01222-8. Epub ahead of print. PMID: 34097852.

6: Khazal S, Kebriaei P. Hematopoietic cell transplantation for acute lymphoblastic leukemia: review of current indications and outcomes. Leuk Lymphoma. 2021 Jun 3:1-14. doi: 10.1080/10428194.2021.1933475. Epub ahead of print. PMID: 34080951.

Author Response

Dear reviewer,

Thank you for your report. Here are my responses to your comments:

  • In the updated version of our article we tried to update the information as to be closer to a complete guide
  • We removed the first table because the information, indeed, it was repetitive
  • We tried best to complete the information regarding the different characteristics of AYA in Table 2 (which now is called Table 1), but still some information is missing from the literature (like the percentage of ETP-ALL, IKZF1 deletion and IKZFplus in AYA)
  • We followed the instructions and put the information about the treatment in another table, together with some new information (such as novel therapies, including blinatumomab, CAR-T cell therapy)
  • We updated some information and bibliography in accordance with the relevant new references from the literature
  • Each table has a separate column for references

Reviewer 2 Report

The Authors presented a review focused on the reasons for the different outcome of children, AYA and older adults affected by ALL.

Overall, the paper is quite comprehensive, clear and well written.

I suggest some edits:

- line 169-172: this information needs to be mitigated as there is controversy about the superiority of kinetics of response over conventional risk factors, including phenotype and genetics

- table 2: specify that the 85% frequency for t(4;11)/MLL forms does not apply to the entire population of children, but only to infants

- lines 293-297: this concept may be better developed and references should be expanded and updated

- lines 399: the role of novel treatments, particularly blinatumomab, in reducing MRD before transplant should be discussed here

- paragraph 6 (future directions in treating ALL): references for this section are a bit outdated (53, 54); overall, the Authors miss to show how these novel agents are being incorporated in frontline treatment of ALL protocols, and this is their main conclusion (lines 458-461), with which I fully agree; however, this subject needs to be addressed in much more detail

Author Response

Dear reviewer,

Thank you for your review. Here are our responses to your comments:

  • in lines 169-172 we tried to best explain that some studies indicate that conventional risk factors could weight much less than other oncogenetic events and early MRD assessment in establishing the risk of relapse
  • we specified in Table 2 (which now is Table 1) that the 85% of patients with MLL rearrangement were infants
  • lines 293-297: we did not discuss the concept in much detail in that paragraph because we addressed it in the subsequent paragraphs
  • line 399: we updated the paragraph about the transplantation and we introduced the concepts of novel therapies as bridges to hematopoietic stem cell transplantation
  • we updated the 6th paragraph with the newest developments in the field and consequently we updated the references. 

Round 2

Reviewer 1 Report

This review displays a convenient and useful comparison between age groups, attending to different parameters. In my opinion, the authors have improved the article so that the information is now more complete. Unoriginal table 1 has been removed, which I think is right; Incomplete table 2 (now table 1) has been improved; the authors added an additional table to gather information on treatments, although its format is not convenient and it should be improved.  The authors added several recent references, although I recommend a few examples either more recent and/or in journals of higher impact, which could be also added.

In version 2, the authors have answered to my specific comments in the following manner:

REVIEWER_1st report: Table 1 contains exactly the same information as figure 2 in (doi: 10.3324/haematol.2015.141101), thus adding no significant knowledge to the field.

AUTHORS: In the updated version of our article we tried to update the information as to be closer to a complete guide. We removed the first table because the information, indeed, it was repetitive.

REVIEWER_2nd report: I agree.

REVIEWER_1st report: Table 2 lacks some information that could be have been obtained from the literature, like JAK alterations in AYAs (doi:10.1056/NEJMoa1403088) or IKZF1 implication in AYAs (doi: 10.1200/JCO.2017.76.7228). Moreover, this table would need an additional column with the referred articles for each row.

AUTHORS: We tried best to complete the information regarding the different characteristics of AYA in Table 2 (which now is called Table 1), but still some information is missing from the literature (like the percentage of ETP-ALL, IKZF1 deletion and IKZFplus in AYA). Each table has a separate column for references.

REVIEWER_2nd report: Despite lacking some information, the table has been improved.

REVIEWER_1st report: Additionally, I miss some other table(s) gathering the information for sections related to treatment (differences, toxicities, response / resistance, etc).

AUTHORS: We followed the instructions and put the information about the treatment in another table, together with some new information (such as novel therapies, including blinatumomab, CAR-T cell therapy). Each table has a separate column for references.

REVIEWER_2nd report: The information gathered in this table is relevant, but the table in its present format is confusing and somewhat difficult to follow. It probably requires a more convenient format with appropriate lines and separations. Moreover, it would benefit from shortening (shorter sentences, just itemizing concepts).

REVIEWER_1st report: The bibliography does not include relevant recent references.

AUTHORS: We updated some information and bibliography in accordance with the relevant new references from the literature.

REVIEWER_2nd report: The authors included appropriate references. However, I still miss some relevant papers regarding section MRD evaluation, as an example:

- Balbach, S.T.; Makarova, O.; Bonn, B.R.; Zimmermann, M.; Rohde, M.; Oschlies, I.; Klapper, W.; Rossig, C.; Burkhardt, B. Proposal of a genetic classifier for risk group stratification in pediatric T-cell lymphoblastic lymphoma reveals differences from adult T-cell lymphoblastic leukemia. Leukemia 2016, 30, 970-973, doi:10.1038/leu.2015.203.

- Beldjord, K.; Chevret, S.; Asnafi, V.; Huguet, F.; Boulland, M.L.; Leguay, T.; Thomas, X.; Cayuela, J.M.; Grardel, N.; Chalandon, Y., et al. Oncogenetics and minimal residual disease are independent outcome predictors in adult patients with acute lymphoblastic leukemia. Blood 2014, 123, 3739-3749, doi:10.1182/blood-2014-01-547695.

- Lepretre, S.; Touzart, A.; Vermeulin, T.; Picquenot, J.M.; Tanguy-Schmidt, A.; Salles, G.; Lamy, T.; Bene, M.C.; Raffoux, E.; Huguet, F., et al. Pediatric-Like Acute Lymphoblastic Leukemia Therapy in Adults With Lymphoblastic Lymphoma: The GRAALL-LYSA LL03 Study. J Clin Oncol 2016, 34, 572-580, doi:10.1200/JCO.2015.61.5385.

Author Response

Dear reviewer,

Thank you very much for your time and effort in reviewing our revised manuscript. We have addressed all the issues raised:

  •  We modified the Table 2 format as to be more easy to read and we summarised the information it contained
  • We updated the references according to your suggestions with relevant articles from the literature regarding MRD and we added a few other new articles.
  • Also, we have edited the manuscript for English language and we hope you will find that it improved in clarity and readability.